# Reverse Vaccinology Approach to Identify Novel and Immunogenic Targets against *Streptococcus gordonii*

**DOI:** 10.3390/biology13070510

**Published:** 2024-07-09

**Authors:** Aneeqa Abid, Badr Alzahrani, Shumaila Naz, Amina Basheer, Syeda Marriam Bakhtiar, Fahad Al-Asmari, Syed Babar Jamal, Muhammad Faheem

**Affiliations:** 1Department of Bioinformatics and Biosciences, Capital University of Science and Technology, Islamabad 44000, Pakistan; aneeqaabid95@gmail.com (A.A.); marriam@cust.edu.pk (S.M.B.); 2Department of Clinical Laboratory Sciences, College of Applied Medical Sciences, Jouf University, Sakaka 72388, Saudi Arabia; baalzahrani@ju.edu.sa; 3Department of Biological Sciences, National University of Medical Sciences, Rawalpindi 46000, Pakistan; shumaila.naz@numspak.edu.pk (S.N.); abasheer.msib08asab@student.nust.edu.pk (A.B.); 4Department of Food and Nutrition Sciences, College of Agricultural and Food Sciences, King Faisal University, Al Ahsa 31982, Saudi Arabia; fahadalasmarifst@kfu.edu.sa; 5Department of Biomedical Sciences, School of Medicine and Health Science, University of North Dakota, Grand Forks, ND 58203, USA

**Keywords:** *Streptococcus gordonii*, infective endocarditis, pan-genomics, subtractive-genomics, docking, TLR2

## Abstract

**Simple Summary:**

This study utilizes subtractive proteomics and pangenome analysis to identify essential non-homologous pathogenic surface proteins that may serve as potential targets for vaccine candidates. The vaccine design incorporates specific B- and T-cell epitopes coupled with adjuvants to enhance the immune response. Computational methods, such as in silico gene cloning, docking studies, and immunological simulation, are employed to assess the efficacy of the vaccine design. The study emphasizes the necessity for further validation through in vitro and in vivo testing to confirm the safety and effectiveness of the vaccine, despite the positive outcomes shown. This method represents a significant advancement in utilizing immunization to proactively prevent infections caused by *S. gordonii*.

**Abstract:**

*Streptococcus gordonii* is a gram-positive, mutualistic bacterium found in the human body. It is found in the oral cavity, upper respiratory tract, and intestines, and presents a serious clinical problem because it can lead to opportunistic infections in individuals with weakened immune systems. Streptococci are the most prevalent inhabitants of oral microbial communities, and are typical oral commensals found in the human oral cavity. These streptococci, along with many other oral microbes, produce multispecies biofilms that can attach to salivary pellicle components and other oral bacteria via adhesin proteins expressed on the cell surface. Antibiotics are effective against this bacterium, but resistance against antibodies is increasing. Therefore, a more effective treatment is needed. Vaccines offer a promising method for preventing this issue. This study generated a multi-epitope vaccine against *Streptococcus gordonii* by targeting the completely sequenced proteomes of five strains. The vaccine targets are identified using a pangenome and subtractive proteomic approach. In the present study, 13 complete strains out of 91 strains of *S. gordonii* are selected. The pangenomics results revealed that out of 2835 pan genes, 1225 are core genes. Out of these 1225 core genes, 643 identified as non-homologous proteins by subtractive proteomics. A total of 20 essential proteins are predicted from non-homologous proteins. Among these 20 essential proteins, only five are identified as surface proteins. The vaccine construct is designed based on selected B- and T-cell epitopes of the antigenic proteins with the help of linkers and adjuvants. The designed vaccine is docked against TLR2. The expression of the protein is determined using in silico gene cloning. Findings concluded that Vaccine I with adjuvant shows higher interactions with TLR2, suggesting that the vaccine has the ability to induce a humoral and cell-mediated response to treat and prevent infection; this makes it promising as a vaccine against infectious diseases caused by *S. gordonii.* Furthermore, validation of the vaccine construct is required by in vitro and in vivo trials to check its actual potency and safety for use to prevent infectious diseases caused by *S. gordonii*.

## 1. Introduction

*Streptococcus gordonii* (*S. gordonii*) is a non-pathogenic bacteria found in the skin, mouth cavity, upper respiratory tract, and intestines. It mostly lives on mucosal surfaces but it also exists in water, soil, plants, and food [1]. *S. gordonii,* a member of the α-hemolytic (viridans) sanguinis group, is mostly found in the oral cavity of humans and animals. However, it is also an opportunistic pathogen, causing various diseases and infections, including infective endocarditis and apical periodontitis [2]. Oral streptococci comprising *S. gordonii (Sg*) and *S. sanguinis* (*Ss*) are one of the most common colonies of biofilms in dental areas, i.e., dental plaque; up to 70% forms in clean dental areas [3]. *Sg* and *Ss* can attach to parts of the salivary pellicle and oral bacteria by using a wide range of adhesion proteins that are expressed in cells [4]. This interaction initiates the formation of dental plaque. Additionally, *S. gordonii* and *S. sanguinis* can invade the bloodstream and are a primary cause of infective endocarditis (IE). These are α-hemolytic oral streptococci and have recently been found in neutropenic infections of blood [5]. It attaches to host cell surfaces and triggers inflammatory responses, contributing to disease development [6].

*S. gordonii* is a causal agent of endocarditis that has varying morbidity and fatality rates in low-, middle-, and high-income nations. South Latin America and Eastern Europe have the highest death rates, followed by East Asia [7]. Over 5 years, there is a 40% death rate, with 22% of cases occurring in hospitals. Between 2011 and 2016, there were 4.4 cases per 100,000 people in rural upstate New York; with 45 confirmed and nine potential cases [8]. *S. gordonii* produces cell wall proteins such as Streptococcal surface protein (Ssp) A, SspB, collagen-binding domain protein (CbdA), and SRR glycoproteins such as GspB. The cell wall proteins can bind to platelets, erythrocytes, monocytes, and dendritic cells, potentially triggering acute immunological responses in humans [9,10].

The treatment for IE varies by age, but generally involves a mix of aminoglycosides and vancomycin. Surgery is recommended for 40%–45% of IE patients due to the multifarious nature of their condition. Antibiotic resistance hinders effective prevention of infectious endocarditis. *S. gordonii* is used as a live oral vaccination to control *Schistosoma japonicum* (*S. japonicum*) worms, expressing the M6-Sj-F1 fusion protein protected mice from *S. japonicum* infections [11,12]. As a result, the most effective alternative treatment is a vaccine, which is estimated to save millions of lives each year [13]. Molecular-omics methods were used to identify immunogenic peptides, including T-cell and B-cell epitopes, and develop a vaccination sequence [14].

Antibiotic treatment is available against infection caused by *S. gordonii*. However, antibiotic treatment is not long-lasting and infection can occur again. Vaccination has a long-lasting impact on the organism to prevent infectious diseases. Therefore, by developing vaccine, one can develop immunity against *S. gordonii* and prevent infections and diseases caused by it, such as infective endocarditis (IE).

The accessibility of a complete parasitic proteome facilitates several computational techniques. Reverse vaccinology and comparative proteomics approaches are successful for enhancing potential targets and recognizing those that are fundamental for normal cell functions, as well as those that are harmful in the host cell. This approach enables us to recognize immunogenic and antigenic vaccine targets that are significant for pathogenesis [15]. A number of linkers and adjuvants were combined with antigenic, non-allergic, and non-toxic T- and B-cell epitopes to enhance immune responses and increase the vaccine’s overall efficacy. However, the study’s methodology concentrated on using strict parameters to provide the most accurate possible predictions; nevertheless, the pipeline employed here had a number of drawbacks [15]. For example, while their reliability has been well demonstrated, predictions regarding the MEV epitopes to MHC alleles and B cells require extensive validation, to ensure that the epitopes are arranged in the correct order to study the immunogenicity of the MEV in an experimental setting.

Furthermore, animal models must validate the developed vaccine in order to verify its efficacy against Schistosoma infections [15]. A computational method, more precisely an immune-informatics technique, was used in this research to build and create a dependable multi-epitope subunit vaccination that may be effective against infectious endocarditis. The four in vivo reported proteins—GspB, CshA, CshB, and SspA—were retrieved in the first stage [16]. Next, CTL, HTL, and B-cell epitopes were selected, and the epitopes were attached to the relevant linkers. Analyzing the vaccine’s antigenic, allergic, toxic, and physiochemical qualities helped to preserve its validity. The vaccination and human TLR2 were docked to find the most stable docked complex with the most hydrogen bonds and salt bridges [16]. To verify its expression, the peptide sequence was optimized, reverse translated, and added to plasmid pET28a + for cloning. Immune response triggered by the constructed multi-epitope subunit vaccine validated its immune simulation. As a result, this study additionally needs practical validation in a wet lab, which is the only limitation of the current study. It will stimulate a long-standing immunity and will help in controlling *S. gordonii-*associated infections [16].

The reverse vaccinology approach (RV) involves reversing pasture vaccinology and using genomic technology to uncover potential antigenic and immunogenic agents in bacterial and virus proteomes; numerous studies have demonstrated the identification and prioritization of vaccine targets for diverse diseases [17,18]. The current study analyzed the bacterial pan-genome to identify essential, accessory, and unique proteins of a certain species. Using the RV, a multi-epitope vaccine (MEV) for *S. gordonii* was designed using sequence-conservation-based features and core proteins that have no sequence resemblance to human proteins. Core genes and non-homologous proteins were identified through subtractive genomics. Essential proteins were predicted from non-homologous proteins. The proteins collected for this study were surface proteins; the antigenic proteins were selected, and the vaccine was designed based on selected B- and T-cell epitopes of the antigenic proteins, with the help of linkers and adjuvants. The designed vaccine was docked against TLR2. TLR2 plays an important role in the human immune system. Then, the expression of the protein was determined using in silico gene cloning, resulting in a potential MEV construct for *S. gordonii.*

## 2. Materials and Methods

### 2.1. Sequence Retrieval and Genome Selection

All the sequences of genes and proteins were obtained from NCBI (https://www.ncbi.nlm.nih.gov/genome/ (accessed on 1 October 2022)). Only strains with complete genome were selected. The core genome of *S. gordonii* was identified using EDGAR 2.3 software (https://bio.tools/edgar_genomics (accessed on 1 October 2022)) by selecting a reference strain, which is “Challis substr. CH1”. A FASTA sequence of all proteins was obtained from NCBI and then they were BLAST against humans to identify homologous and non-homologous proteins. These non-host homologous proteins were inserted into the DEG database of essential genes (http://origin.tubic.org/deg/public/index.php (accessed on 1 October 2022)) to find out the essential proteins by using thresholds.

### 2.2. Collection of Surface Protein

The subcellular localization of the proteins was identified using the CELLO tool (http://cello.life.nctu.edu.tw/ (accessed on 1 October 2022)). Five surface proteins were present in *S. gordonii* and were selected, including 2 extracellular proteins: YSIRK_signal domain protein and Peptidase C51 domain-containing protein; and 3 membrane proteins: AraC family transcriptional regulator, Glycosyl transferase, and peptidoglycan D, D-transpeptidase FtsI. The sequences were retrieved using Uniprot (https://www.uniprot.org/ (accessed on 1 October 2022)) and virulence was identified using VFDB. The cell wall of *S. gordonii* is composed of lipoproteins, lipoteichoic acids, repetitive serine-rich adhesins, peptidoglycans, and cell wall proteins that are characterized by individual host receptors [1]. They are involved in virulence and immune regulatory processes that induce inflammatory responses. Dimeric receptors containing TLR2 and TLRx recognize the lipoproteins and LTA. SRR adhesins are important for the binding *of S. gordonii* to host cells through sialylated glycans. Nucleotide oligomerization domain (NOD), an intracellular receptor, recognized peptidoglycans. Thus, cell wall elements of *S.gordonii* act as virulence factors that progressively develop diseases with a strong participatory response. These virulence factors often include proteins that are potential candidates for vaccine development because they can provoke a potent immune response [1]. The major virulence factors, such as lipoprotein of *S.gordonii*, are directly recognized by heterodimers; these are composed of toll-like receptors TLR2 along with TLR1 or TLR6 on host cell, including dental pulp cells, dendritic cells, valve interstitial cells, and macrophages. Following activation of TLR2—an adaptor molecule of TLR2—myeloid differentiation primary response 88 (MyD88) mediates the activation of transcription factor that is nuclear factor-kappa B (NF-κB), resulting in the production of pro-inflammatory cytokines and chemokines, maturation of cell, and infiltration of immune cells into lesions. These processes are involved in inducing inflammatory responses and thus result in development of diseases such as apical periodontitis or infective endocarditis [1].

Molecular weight of proteins was calculated using the ProtParam tool (https://web.expasy.org/protparam/ (accessed on 1 October 2022)). VaxiJen v2.0 webserver (http://www.ddg-pharmfac.net/vaxijen/VaxiJen/VaxiJen.html (accessed on 1 October 2022)) at threshold 0.4 was used to predict antigenicity. The allergenicity of the proteins was predicted through AllerTOP v.2.0 (https://www.ddg-pharmfac.net/AllerTOP/ (accessed on 1 October 2022)). The physiochemical properties of the selected proteins were measured using ProtParam (https://web.expasy.org/protparam/ (accessed on 1 October 2022)). The solubility of the protein was predicted through SoluProt (https://loschmidt.chemi.muni.cz/soluprot/ (accessed on 1 October 2022)). To predict disulfide bonds in both proteins, DIANNA 1.1 (http://clavius.bc.edu/~clotelab/DiANNA/ (accessed on 1 October 2022)) was used. The function and pathway analysis of selected proteins was done through uniprot (https://www.uniprot.org/ (accessed on 1 October 2022)) by using uniprot ID of the protein. The pathway of the protein was also predicted using the KEGG web tool (https://www.genome.jp/kegg/kegg4.html (accessed on 1 October 2022)).

### 2.3. Epitope Prediction

One of the important steps in an immunoinformatics study is the epitope selection. Multi-specific and broad-based epitopes should be selected. Factors that influence epitope selection include the ability of the epitope to attach to the appropriate MHC molecule, cellular presentation ability of the epitope; and the repertoire of T cells should possess the ability to differentiate between MHC-epitope complexes [19].

### 2.4. Prediction of B-Cell Epitopes

B-lymphocytes epitopes were predicted using the online webserver ABCpred (https://webs.iiitd.edu.in/raghava/abcpred/ABC_submission.html (accessed on 1 October 2022)). The fasta sequences of proteins were entered in the ABCpred at threshold 0.5 and length of epitope is 16 [20]. After the prediction of B-cell epitopes, VaxiJen v2.0 tool at threshold of 0.4 (http://www.ddg-pharmfac.net/vaxijen/VaxiJen/VaxiJen.html (accessed on 1 October 2022)) and AllerTOP v2.0 tool (https://www.ddg-pharmfac.net/AllerTOP/ (accessed on 1 October 2022)) was used for the prediction of the antigenicity and allergicity of the epitopes (Sana et al., 2022) [20]. The epitopes which were antigenic and non-allergenic were selected. The toxicity of the epitopes was predicted using the ToxinPred tool (http://crdd.osdd.net/raghava/toxinpred/multi_submit.php (accessed on 1 October 2022)), and the sub-cellular localization of epitopes was also predicted using the TMHMM-2.0 web server (https://services.healthtech.dtu.dk/service.php?TMHMM-2.0 (accessed on 1 October 2022)).

### 2.5. Prediction of T-Cell Epitope

A T-cell epitope prediction tool of IEDB analysis resource (http://tools.iedb.org/main/tcell/ (accessed on 1 October 2022)) was used to evaluate T-cell epitopes. This tool determines the binding affinity of peptides towards the MHC-I and MHC-II on the basis of their IC50 value. On the basis of the IC50 score, MHC-I and MHC-II epitopes were selected.

For MHC-I binding epitopes (CTL), the prediction method selected was ANN 4.0 and the MHC resource species was human. All the HLA-A* alleles were selected and the predicted length of the epitope was 9-mer (Priyadarshini et al., 2014) [21].

For MHC-II-restricted epitopes (HTL) the prediction method selected was NN-align 2.3 and the MHC resource species was human, and locus was HLA-DR. All the HLA-DRB1* alleles were selected and the predicted length of epitope was 15 [21,22].

VaxiJen v2.0 and AllerTOP v2.0 was used to determine the peptides’ antigenicity and allergenicity. The sub-cellular localization of non-allergenic epitopes were determined using TMHMM-2.0 [23] and ToxinPred (http://crdd.osdd.net/raghava/toxinpred/multi_submit.php (accessed on 1 October 2022)) was employed to determine the toxicity of the selected epitope and other physiochemical properties including hydrophobicity, hydrophilicity, charge, theoretical isoelectric point value (PI). The ProtParam server (https://web.expasy.org/protparam/ (accessed on 1 October 2022)) was used to determine the molecular weight of the non-toxin epitopes [24].

### 2.6. Epitope Conservation Analysis

The epitope conservancy analysis of B-cells, MHC class-I and MHC class-II alleles was performed by using the IEDB conservancy tool (http://tools.iedb.org/conservancy/ (accessed on 1 October 2022)). This will predict the percentage of sequence matches at identity greater than or equal to 100%.

### 2.7. Population Coverage Prediction of T-Cell Epitopes

The HLA genotype frequencies among different populations of the world vary (Sana et al., 2022) [20]. Differences in transmission and expression of the HLA alleles helps in designing an epitope-based vaccine. Thus, the binding ability of a specific epitope selected for the multivalent vaccine to HLA-I and II molecules is affected by HLA polymorphism [25].

### 2.8. Construction of Multivalent Vaccine Design

On the basis of high antigenicity, high binding affinity (low IC50 value) and non-allergenic nature, a set of B-cell (LBL), and T-cell including CTL (MHC-I) and HTL (MHC-II) epitopes were selected. The multivalent vaccine was designed by joining the adjuvant, epitopes of T-(HTL and CTL) and B-cell with their respective linkers. The selected epitopes were linked together with the help of linkers for designing a multi-vaccine construct. The linkers used in multivalent vaccine construct were EAAK, GPGPG, and AAY (Sana et al., 2022) [20]. The addition of an adjuvant at the N-terminus of vaccine construct with the help of EAAK linker. After the adjuvant, B-cell epitopes were linked by KK linker. B-cell epitopes were then linked to HTL (MHC-II epitopes) by GPGPG linker. HTLs were connected with each other by GPGPG linker. To link HTL with CTL (MHC-I epitopes), an AAY linker was used. Furthermore, CTLs were connected with each other by AAY linker. At last, the His-tag (six histamines) was added at the C-terminus of the vaccine to finish the joining of the vaccine final construct [26]. The vaccine antigenicity and allergenicity was predicted through the VaxiJen v2.0 and AllerTOP v2.0 server using the ProtParam tool to determine the physiochemical properties and molecular weight (MW) of the vaccine, its theoretical isoelectric point value (PI), its instability index (II), its aliphatic index; and GRAVY SOLUPROT v1.0 (https://loschmidt.chemi.muni.cz/soluprot/ (accessed on 1 October 2022)) is used to find the solubility of the vaccine construct. This tool predicts vaccine expression in organism *E. coli*. If the vaccine solubility is above 0.5, the protein will have good expression in humans.

### 2.9. Structure Prediction, Refinement, and Validation of Multi-epitope Vaccine

The PSIPRED server (http://bioinf.cs.ucl.ac.uk/psipred/&psipred_uuid=7d99d650-1289-11ed-acee-00163e100d53 (accessed on 1 October 2022)) was used for the secondary structure prediction of vaccine construct. Protein primary sequence is submitted in this tool by selecting predict secondary structure. It will predict the protein secondary structure in the form of coils, helix, and sheets. This server shows the results with high accuracy.

The tertiary structure of vaccine was predicted using the iTASSER server (https://seq2fun.dcmb.med.umich.edu//I-TASSER/ (accessed on 1 October 2022)). This tool generated a 3D model of protein from the sequence of its amino acids. To predict the accuracy of the protein model, iTASSER also provides a confidence score. Pymol software v2.4. is used to view and check the 3D (tertiary) structure. The predicted tertiary structure model was refined by using GalaxyRefine (https://galaxy.seoklab.org/cgi-bin/submit.cgi?type=REFINE (accessed on 1 October 2022)) and trRosetta (https://yanglab.nankai.edu.cn/trRosetta/ (accessed on 1 October 2022)). These tools improved the global and structural quality of a tertiary structure. The quality of protein tertiary structures was refined and improved by GalaxyRefine and trRosetta, especially the protein structure prediction. Both these tools improved the global and structural quality of tertiary protein structures by refining erroneousness, decreasing steric clashes, and improving local interactions. These tools are useful for increasing the accuracy of predicted protein structures; this is beneficial in many applications such as structural biology, drug discovery, and protein engineering [27].

### 2.10. Validation of Vaccine 3D Structure

The additional validation of the protein vaccine structure was obtained through Ramachandran plot by using a Ramachandaran plot server (https://zlab.umassmed.edu/bu/rama/index.pl (accessed on 1 October 2022)). The percentage of error comparisons of predicted structure residues of the refined protein compared to unrefined protein was predicted using the ERRAT server (https://saves.mbi.ucla.edu/ (accessed on 1 October 2022)). The Z-score of refined structure was predicted through the molprobity server (http://molprobity.biochem.duke.edu/index.php?MolProbSID=s5mqsj74op6r8o3u1j5jkeasr2&eventID=64 (accessed on 1 October 2022)). NetChop-3.1 (https://services.healthtech.dtu.dk/service.php?NetChop-3.1 (accessed on 1 October 2022)) at threshold 0.5 was used for the prediction of sites of proteasomal cleavages. The pathway processing of MHC-I ligands was mostly composed of two steps: (i) initially, proteasome degrades the protein; (ii) the degraded products were transported along with antigen processing (TAP) to the endoplasmic reticulum via associated transporter [28].

### 2.11. Molecular Docking with TLR2

The immunologic receptors, including toll-like receptors (TLRs) are necessary for electing a response from immune system against infection caused by pathogenic organisms. During infection caused by *S. gordonii,* the TLR involved in evoking immune response in humans is TLR2 protein-peptide molecular docking technique was used to determine the best binding mode of the multivalent vaccine construct to TLR2 [29]. The RCSB database (https://www.rcsb.org/ (accessed on 1 October 2022)) retrieves the protein sequence of TLR2. After retrieving the protein sequence, docking of the ligand and receptor was done. ClusPro server (https://cluspro.bu.edu/login.php?redir=/home.php (accessed on 1 October 2022)) was used to carry out docking on TLR2 (receptor) and vaccine construct (ligand). The interactions between the ligand (vaccine) and receptor was determined through PDBePISA (https://www.ebi.ac.uk/pdbe/pisa/ (accessed on 1 October 2022)).

### 2.12. Immune-Simulation

C-IMMSIM, an online webserver was used for immune simulation of vaccine construct. It simulates the response of a host immune system to the multivalent vaccine design construct. The server is based on modeling approach and estimated the effect caused by foreign particle or antigen on the immune system using PSSM method. C-IMMSIM calculated the production of cytokines, interferon, and antibodies after the injection of vaccine.

### 2.13. Gene Cloning

EMBOSS Backtranseq webserver (https://www.ebi.ac.uk/Tools/st/emboss_backtranseq/ (accessed on 1 October 2022)) was used to translate the protein of the vaccine (amino acids to nucleotides). The server result shows that the protein was translated to nucleotides. The vaccine nucleotide sequence from EMBOSS Backtranseq was entered into JCat (http://www.jcat.de/ (accessed on 1 October 2022)) to adapt codon usage of vaccine (Sana et al., 2022) [20]. In codon-adaptation, the optimized DNA sequence was obtained by selecting *E. coli* (Strain K12) as an organism. After the optimization of codon, the multivalent vaccine polypeptide with improved DNA sequence was obtained. The improved DNA sequence predicted from process of codon adaptation was subjected to in silico cloning through SnapGene tool (https://www.snapgene.com/ (accessed on 1 October 2022)). The improved codon has been inserted into multiple cloning sites (MCS) of pET-28a (+) of the *E. coli* vector) [30].

## 3. Results and Discussion

### 3.1. Sequence Retrieval and Genome Selection

Genome Assembly and Annotation from NCBI showed a total of 91 strains of *S. gordonii.* Out of 91 strains of *S. gordonii,* 13 are complete, 65 are contig, and 13 are scaffold. The proteins and genes of the 13 complete genomes of *S. gordonii* were retrieved from NCBI and were included in this study. The EDGAR software v3.0. requires selection of reference strain. The reference strain selected was “Challis substr. CH1” based on its release date that was compared with other 12 strains. The total number of genes identified in pan-genome is 2835 genes; out of these, 1225 were core genes. From 1255 core genes, 643 were identified as non-homologous proteins. The identified non-host homologous proteins were used to identify essential proteins by inserting the proteins into the DEG database of essential genes to find out the essential proteins by using thresholds. The default parameters will be selected: E value = 0.0001, bit score ≥ 100, and identity ≥ 25%. A total of 20 proteins were attained.

### 3.2. Collection of Surface Proteins

Five surface proteins were selected, including two extracellular proteins (YSIRK_signal domain protein, and peptidase C51 domain-containing protein) and three membrane proteins (AraC family transcriptional sregulator, Glycosyl transferase, peptidoglycan D, D-transpeptidase FtsI) using the CELLO tool. The amino acids sequence of all the five surface proteins was retrieved from Uniprot using uniprot id of the protein i.e., A0A0A6S1K7, Q9AB82, F0HZA2, Q9AEU1, D0CCM7. The gene name was also predicted from uniprot. Virulent proteins were predicted through VFDB. VFDB shows that all the five targeted surface proteins are virulent. Virulent proteins reveal novel therapeutics targets. The ProtParam tool calculated the molecular weight of the proteins. VaxiJen v2.0 results showed that only two proteins possess antigenic properties; namely, YSIRK_signal domain protein and peptidoglycan D,D-transpeptidase FtsI, and these two proteins were selected for further analysis. The AllerTOP server shows that all the five proteins were non-allergenic. The details are given in Table 1.

Then, the proteins were selected on the basis of antigenicity and allergicity. YSIRK_signal domain protein and peptidoglycan D,D-transpeptidase FtsI were antigenic and non-allergenic, so these two proteins were selected for further for further analysis.

Structural analysis of YSIRK_signal domain protein and peptidoglycan D, D-transpeptidase FtsI was done by screening the physio-chemical properties of the antigenic proteins using the ProtParam tool (Expasy). YSIRK_signal domain protein has 469 amino acids with a molecular weight of 53,992.06 kDa, while peptidoglycan D, D-transpeptidase FtsI has 610 amino acids with a molecular weight of 67,659.05 kDa. The theoretical pI value shows the nature of the protein. If the theoretical pI value of the protein is below 7, the nature of protein will be negative and if the theoretical pI value of the protein is above 7, the nature of the protein will be positive. The theoretical pI values of both proteins were above 7, so the nature of both proteins was positive. Other physiochemical properties including instability index, aliphatic index, gravy, half-life, atomic composition, the total number of positively charged residues (Arg + Lys) and the total number of negatively charged residues (Asp + Glu), were also predicted using ProtParam and shown in the Table 2. The SoluProt was used to predict solubility of the proteins. The result from SoluProt shows YSIRK_signal domain protein 0.547 and peptidoglycan D, D-transpeptidase FtsI 0.537. DIANNA 1.1 was used to predict the disulfide bond in both proteins. The result showed that there was no disulfide bond in YSIRK_signal domain protein neither in peptidoglycan D, D-transpeptidase FtsI.

The function of the protein includes its molecular and biological function (Table 3). The function of peptidoglycan D, D-transpeptidase FtsI is to catalyze cross-linking of the peptidoglycan cell wall at the division septum. The KEGG web tool shows the pathway of both proteins (Table 3).

### 3.3. B-cell Epitope Prediction and Selection

Selected length of epitope was 16 at threshold 0.5. Subsequently, 45 epitopes were predicted for YSIRK_signal domain-containing protein and 65 epitopes were predicted for peptidoglycan D, D-transpeptidase FtsI. VaxiJen v2.0, AllerTOP v2.0, ProtParam, and ToxinPred tools were used to predict antigenicity, allergenicity, molecular weight (MW), and toxicity of the epitopes, respectively.

The necessary peptides were selected based on high antigenic value. The peptides that were non-allergenic, non-toxic, present outside, and have high antigenic value were selected; one epitope from YSIRK_signal domain protein was selected and two epitopes from peptidoglycan D, D-transpeptidase FtsI were selected. The details are given in Table 4.

### 3.4. T-Cell Epitope Selection

MHC-I restricted epitopes obtained using IEDB. All HLA-A* alleles were obtained. Approximately, 5520 alleles for 335 common peptides of YSIRK_signal domain-containing protein and 7200 alleles for 450 common peptides of peptidoglycan D, D-transpeptidase FtsI were obtained. The significant epitopes from common epitopes were selected on the basis of low IC50 value of the predicted peptides. The epitope was selected on the basis of certain properties including determination of antigenicity, allergicity, trans-membrane helicase, and toxicity. The peptides with antigenicity above 0.4 were selected using VaxiJen v2.0. The results are shown in Table 5.

Antigenic analysis is an important step in designing vaccines. The epitopes with an antigenic value above 0.4 were selected for further analysis through VaxiJen v2.0. An online prediction server was used to make sure that the vaccine does not cause any allergic reaction. The epitopes that were non-allergenic were selected through AllerTOP v2.0, and other allergen epitopes were discarded. The epitopes that were present outside were selected through TMHMM-2.0 and others will be discarded. Non-toxin peptides were selected through ToxinPred. After applying the above tool of prediction, only 37 peptides out of 335 peptides of YSIRK_signal domain-containing protein and 54 peptides out of 450 peptides of peptidoglycan D,D-transpeptidase FtsI were left that were antigenic, non-allergen, non-toxin and present outside. The epitopes were arranged according to their low to high IC50 value. Three epitopes of YSIRK_signal domain-containing protein and four epitopes of peptidoglycan D,D-transpeptidase FtsI were selected on the basis of their properties, as shown in Appendix A.

Hydrophobicity, hydrophilicity, charge, and pI value of the selected epitopes were predicted through ToxinPred. The molecular weight (MW) of epitopes was predicted through Potparam. Another vital feature is the prediction of peptide-digesting enzymes using the peptide cutter tool. If peptides are digested by fewer enzymes, they are considered to be stable peptides and are more favorable vaccine targets, whereas the peptides that are digested by several enzymes are considered to be non-stable. The result from the peptide cutter tool shows that the peptides are stable because these are digested by a small number of enzymes. Appendix A.

MHC-II restricted epitopes were obtained using IEDB. The prediction method selected was NN-align 2.3, the MHC resource species was human, and the locus was HLA-DR. All the HLA-DRB1* alleles were obtained and the predicted length of epitope was 15. Approximately 4921 alleles for 257 common peptides of YSIRK_signal domain-containing protein and 6270 alleles for 300 common peptides of peptidoglycan D,D-transpeptidase FtsI were obtained. The significant epitopes from common epitopes were selected on the basis of a low IC50 value of the predicted peptides. The epitope was selected on the basis of certain properties, including determination of antigenicity, allergicity, trans-membrane helicase, and toxicity. VaxiJen v2.0 is used to predict antigenicity of epitopes. The peptides with antigenicity above 0.4 were selected. The results are shown in Table 6.

The necessary peptides were selected based on their high antigenic value. Antigenicity analysis is important step in designing a vaccine. VaxiJen v2.0 generates the antigenic score of epitopes. The AllerTOP v2.0 tool predicts allergenicity and epitopes that were non-allergenic were selected, and other allergen epitopes were discarded. TMHMM-2.0 predicts epitopes that were present outside and were selected, and others will be discarded. ToxinPred predicted the toxicity of the epitopes that were present outside. Non-toxin peptides will be selected. After applying the above properties only 31 peptides out of 257 peptides of YSIRK_signal domain-containing protein and 39 peptides out of 300 peptides of peptidoglycan D, D-transpeptidase FtsI were left that were antigenic, non-allergen, non-toxin, and present outside. The epitopes that were antigenic, non-allergenic, non-toxin, present outside, and having low IC50 value were selected. Three epitopes of YSIRK_signal domain-containing protein and four epitopes of peptidoglycan D, D-transpeptidase FtsI were selected on the basis of the above properties (Table 7).

### 3.5. Determination of Physiochemical Properties and Vital Features

Hydrophobicity, hydrophilicity, charge, and pI value were predicted through ToxinPred. Molecular weight of epitopes was predicted through ProtParam. Another vital feature is the prediction of peptide-digesting enzymes by using the peptide cutter tool. If peptides are digested by fewer enzymes, they are considered to be stable peptides and are more favorable vaccine targets, whereas the peptides that are digested by several enzymes are considered to be non-stable. The result from the peptide cutter tool shows that the peptides are stable because these are digested by a small number of enzymes. The physio-chemical properties of MHC-II restricted epitopes are given in Appendix A.

### 3.6. Epitope Conservation Analysis

The IEDB conservancy tool performs the epitope conservation analysis of B-cells, MHC Class-I, and MHC class-II alleles, as shown in Appendix A.

### 3.7. Multivalent Vaccine Design Construction

The sequence of final vaccine composed of a total 410 amino acids with three overall B-cell epitopes comes after seven MHC-II (HTL) epitopes and seven MHC-I (CTL) epitopes. The epitopes were detached by their defined linkers. Cholera enterotoxin subunit B was attached as an adjuvant at the N-terminus of the vaccine construct for increasing the immunogenicity; and at C-terminus of the vaccine, six histidine tag were added. An EAAK linker is used as a rigid linker in the multivalent vaccine. One of the main advantages of the use of linkers is that it maintains its specific functions from epitopes by preserving the structured separation of its functional domains. B-cell epitopes were linked by KK linker (Lysine linker). The KK linker was selected mainly because of three reasons: (i) immunogenic responses was enhanced by KK linkers; (ii) in the MHC-II restricted antigen presentation, lysosomal proteases such as Cathepsin B are involved in the peptide presentation on the surface of the cell through antigen processing of the peptides, and these lysosomal proteases target the KK linker; (iii) the junctional immunogenicity was reduced through avoiding the antibodies’ induction. For the attachment of the HLA-II epitopes with each other, a GPGPG linker that is a universal spacer was used. The study shows that performed GPGPG linker has ability of inducing TH lymphocyte (HTL) and has the capability to split the epitopes through junctional immunogenicity that restores the immunogenicity of the individual epitopes. Subsequently, AAY (Ala-Ala-Tyr) linker was used to link MHC-I restricted epitopes. In mammalian cells it works as a cleavage site for the proteasomes, so an AAY linker was used to maintain essential junctional immunogenicity between the HLA-I epitope. Immunogenicity of the multi-epitope vaccine was also enhanced by AAY linker. The sequence and order of epitopes in multivalent vaccine construct is shown below (Figure 1 and Figure 2). Moreover, there are some challenges in developing vaccines, including the identification of the appropriate antigens that reduce the chance of autoimmunity and induce a protective immune response against *S. gordonii.* To develop a vaccine that effectually targets *S. gordonii* inside the biofilm is also challenging. For developing a potent vaccine, it is important to understand and control the immunomodulatory effects of *S. gordonii.* Other limitations and challenges for clinical application of the vaccine includes the need for more effective data, the risk of developing autoimmune diseases, and additional safety and care for its delivery.

### 3.8. Antigenicity and Allergenicity Prediction of Multivalent Vaccine Construct

The antigenic score of the protein sequence of vaccine construct with adjuvant and EAAK linker generated by Vaxijen v2.0 was 0.8562, whereas the vaccine without adjuvant and EAAK linker was 1.0571. The antigenic value of the vaccine without adjuvant and EAAK linker is high than the vaccine with adjuvant and EAAK linker. This result shows that both the vaccine proteins were antigenic. This means both the vaccines have the ability to induce an immune response when injected into the human body.

AllerTOP v2.0 server predicts the allergenic potential of the vaccine. The result obtained from this tool shows that the vaccine construct with adjuvant and EAAK linker and without adjuvant and EAAK linker both were non-allergenic and safe to use.

### 3.9. Physicochemical Properties and Solubility Determination of Primary Structure of Vaccine Construct

Physicochemical properties of the vaccine construct, including its molecular weight, theoretical isoelectric point value (PI), instability index, aliphatic index, and GRAVY (grand average of hydropathicity) were evaluated through ProtParam. The result of physiochemical properties obtained from ProtParam shows the molecular weight of protein with adjuvant and EAAK linker was 44,750.94 kDa, and of protein without adjuvant and EAAK linker was 30,412.33 kDa. The theoretical pI value shows the nature of the protein. If the theoretical pI value of the protein is below 7, the nature of protein will be negative and if the theoretical pI value of the protein is above 7, the nature of protein will be positive. The theoretical pI value of vaccine construct with adjuvant and EAAK linker was 7.12, so it is positive whereas theoretical pI value of vaccine construct without adjuvant and EAAK linker was 6.07, so it is negative. The instability index is used to determine whether the protein will be stable in a test tube. If the stability index of protein is less than 40, the protein will be stable in test tube and if the stability index of protein is above 40, the protein will not be stable. The instability index of vaccine protein with adjuvant and EAAK linker was 36.80, so it is stable; and of protein without adjuvant and EAAK linker it was 38.88, so it is also stable.

The aliphatic index of protein with adjuvant and EAAK linker was 88.78, and of protein without adjuvant and EAAK linker it was 88.55. The GRAVY value of a protein determines its hydrophobicity or hydrophilicity. If the GRAVY score of a protein is below 0, it is considered as hydrophilic and if the GRAVY score of a protein is above 0, it is considered as hydrophobic. The GRAVY score of vaccine protein with adjuvant and EAAK linker was 0.135 of protein and GRAVY score of protein without adjuvant and EAAK linker was 0.246. Both the scores were above 0, so both the vaccine proteins were considered as hydrophobic.

The estimated half-life of vaccine protein with adjuvant and EAAK linker is 30 h in mammalian reticulocytes if it tested in vitro; whereas, if tested in vivo it is greater than 20 h in yeast and greater than 10 h in Escherichia coli. The estimated half-life of vaccine protein without adjuvant and EAAK linker is 5.5 h in mammalian reticulocytes if it is tested in vitro; whereas, if tested in vivo it is 3 min in yeast and 2 min in Escherichia coli.

The solubility of vaccine construct obtained from SOLUPROT v1.0 predicts the expression of vaccine in *E. coli*. For the high expression of protein, the solubility should be above 0.5. If the solubility of protein is below 0.5, the protein will be insoluble. The protein solubility value of multivalent vaccine construct of protein with adjuvant and EAAK linker is 0.837; whereas, the protein solubility value of multivalent vaccine construct of protein without adjuvant and EAAK linker is 0.289. The result indicates that the protein sequence of vaccine with adjuvant and EAAK linker is soluble, while that without adjuvant and EAAK linker is insoluble (Appendix A).

### 3.10. Structure Prediction, Refinement, and Validation of Multi-Epitope Vaccine

The vaccine secondary structure was determined by PSIPRED tool (Figure 3). Yellow color indicates the strand, pink color indicates the alpha helix, and grey color indicates coil. Vaccine protein secondary structure shows that it is stable.

### 3.11. Tertiary Structure Prediction

The iTASSER tool initiates the top five final models of the multivalent vaccine construct. Confidence score (C-score) describes the quality of every predicted model [15]. Therefore, the top C-score shows higher confidence in models that were generated. Besides the C-score, iTASSER also calculated TM-score and RMSD value for all the five predicted models of multivalent vaccine design construct, which predicts the overall quality of the model. In this study, the C-score of the first model tertiary structure is higher than all four other models. The C-score for model 1 is −0.41. The estimated TM-score for model-1 is 0.66 and estimated RMSD is 8. Model-1 was therefore selected.

The predicted tertiary structure model from the iTASSER was refined by using GalaxyRefine [16] and trRosetta. The GalaxyRefine tool, based on refinement method when used for refining the selected model-1 generated through iTASSER prediction server by improving both global and local structure quality on average. GalaxyRefine provides five additional models generated by relaxation simulations of the iTASSER model. Model 1 was selected. trRosetta tool refine the tertiary structure predicted model from iTASSER. Transform-restrained Rosetta (trRosetta) server is a web-based platform used for speedy and precise protein structure prediction. It generates five models of the iTASSER model. For the multivalent vaccine designed protein, if the predicted structure models by trRosetta are in low confidence (e.g., having estimated TM-score <0.5), it is possible that this design is not foldable, and wet-lab experiments are not required. The estimated TM-score for model-1 is 0.136 (<0.5). So model-1 was selected.

Ramachandaran plot server validated the structures of vaccine obtained from trRosetta and GalaxyWeb. Results obtained from Ramachandaran plot server for the GalaxyWeb model shows that highly preferred observations were 96.707%, preferred observations were 3.293%, and questionable observations were 0.000% (Figure 4). Therefore, model-1 of GalaxyWeb was selected for docking. The ERRAT value was also evaluated [24]. The ERRAT value before refinement, i.e., the ERRAT value of model-1 of iTASSER was 90.452 (Figure 5) and the ERRAT value after refinement, i.e., ERRAT value of model-1 of GalaxyWeb was 84.085 (Figure 6). The Z-score of the refined structure of (GalaxyWeb) predicted using the molprobity server was 2.05 ± 0.36. The NetChop-3.1 predicts that in the vaccine protein sequence with 410 amino acids polypeptide, the number of cleavage sites was 132.

### 3.12. Molecular Docking of Vaccine Protein with TLR2 and Its Structural Stability

The binding affinity between ligand and receptor molecules is assessed by molecular docking, which is an in silico technique. In this study, Human toll-like receptor 2 (TLR2) was used to predict the binding affinity between the human TLRs and vaccine construct. The 3D structure of TLR2 retrieved from the Protein Data Bank (PDB) (PDB ID: 2z7x). The docking was done by use sssof ClusPro. In ClusPro, the vaccine pdb structure retrieved from GalaxyWeb was uploaded in ligand and 2z7x (TLR2) receptor was uploaded in receptor option and then selected docked. The ClusPro generates 10 models of docking of vaccine with TLR2 (Figure 7). The model 0 was selected on the basis of energy level.

The interactions between the ligand (vaccine) and receptor (2z7x) was determined through PDBePISA. The result from the tool indicates that Vaccine I with adjuvant shows better interactions (Table 8 and Table 9).

### 3.13. Immune Simulation

The immunogenic reaction of the multivalent vaccine design construct was stimulated using the C-ImmSim tool. The vaccine’s immunological response was predicted both with and without the adjuvant and EAAK linker. C-IMMSIM calculated the production of cytokines, interferon, and antibodies after vaccine injection (Figure 8 and Figure 9).

The result shows that the vaccine with adjuvant showed a better immunogenic response. The black line in the graph represent antigens. When the vaccine is injected, it remains in the body for 5 days, as shown in the graph. After 5 days, the body produces antibodies against the vaccine. The value of antibodies is very high in the body, as shown in Figure 8a by brown (IgM + IgG), green (IgM), and purple (IgG1) line in graph. Therefore, the high value of antibodies shows that the vaccine response is very good and this will remain in the body for 32 days. It means the vaccine can induce long-lasting immunity in the body.

### 3.14. Gene Cloning

The protein sequence of the vaccine was translated through EMBOSS. The result from the server shows that the protein was translated to 1230 nucleotides. The codons used in the vaccine structure were matched to codons of *E. coli* (K12) using the codon adaptation tool, JCat. The improved DNA sequence shows the GC content was 53.41% and the CAI score was 0.956%. For the expression analysis of vaccine protein, the protein sequence of vaccine was translated. The EMBOSS Backtranseq webserver was used to translate the protein sequence of the vaccine with adjuvant and EAAK linker. The result from the server shows that the protein (410 amino acids) was translated to 1230 nucleotides.

After the translation of vaccine protein, codon adaptation was performed to improve the DNA sequence of the vaccine. The codons used in the vaccine structure were matched to codons of *E. coli* (K12) using codon adaptation tool, JCat. The improved DNA sequence shows the GC content was 53.41% and the CAI score was 0.956%. (Figure 10).

The length of improved codon sequence obtained from JCat was 1230bp. This improved codon has been inserted into multiple cloning sites (MCS) of pET-28a (+) of the *E. coli* vector between the restriction sites BssSI (3665) and PciI (3224). The final cloned product obtained from cloning is 5637bp long (5637kbp). The length of improved codon sequence obtained from JCat was 1230bp with improved GC content and CAI score. This improved codon has been inserted into multiple cloning sites (MCS) of *E. coli* vector. The pET-28a (+) vector of *E. coli* was used, that is 5369bp long. The improved vaccine codon was inserted between the restriction sites BssSI (3665) and PciI (3224). The final cloned product obtained from cloning is 5637bp long (5.637kbp). This process was the in silico gene cloning of codon optimized vaccine with the expression system *E. coli* K12; pET-28a(+). The vaccine clone was obtained by integrating the vaccine fragment into vector pET-28a (+) of the *E. coli* vector between the restriction sites BssSI (3665) and PciI (3224) (Figure 11).

By employing subtractive proteomics and pangenome analysis, this study finds essential non-homologous virulent surface proteins that could be potential targets for vaccine candidates. Selected B- and T-cell epitopes associated with adjuvants are incorporated into the vaccine construct to improve the immune response. Computational techniques such as in silico gene cloning, docking studies, and immune simulation are used to verify the potential of the vaccine construct. The study highlights the need for additional in vitro and in vivo validation to verify the vaccine’s safety and efficacy, despite its encouraging results. This strategy is a major step toward using vaccination to prevent infections caused by *S. gordonii*.

## 4. Conclusions

The present scientific study suggested a reverse vaccinology (RV) approach toward designing a multi-epitope vaccine using sequence-conservation-based features, and core surface proteins were selected. The two surface proteins were selected according to chosen criteria, i.e., antigenicity above 0.4, non-allergenicity. The antigenic and non-allergenic B-cell and T-cell epitopes selected from the proteins were used to construct a multivalent vaccine with the help of linkers and adjuvants. Vaccine I is constructed with an EAAK linker and adjuvant, whereas Vaccine II is without EAAK linker and adjuvant. Antigenicity and allergenicity of both vaccine suggest that both the vaccines were antigenic and non-allergenic. The physicochemical properties and tertiary structure of the protein were predicted. These predictions were validated and refined. The refined 3D structure of protein was docked with TLR2. The best docked structure was selected, and interaction was predicted. The expression and stability of the protein is determined using in silico gene cloning by inserting the reversely translated and optimized protein sequence into vector pET-28a (+) of the *E. coli*. Findings concluded that Vaccine I with adjuvant shows higher interactions with TLR2, which suggests that Vaccine I has the ability to induce humoral and cell-mediated responses to treat and to prevent infection caused by *S. gordonii*. In future, we can check the experimental validity of our proposed vaccine by the expression and purification of the MEV. For that purpose, we can clone the vaccine construct into an appropriate expression vector and induce expression of the vaccine in the host system and then purification of the expressed protein using techniques such as size exclusion chromatography. The in vitro safety and immunogenicity testing of purified vaccine can be done by number of cytotoxicity assays and immunogenicity assays. Furthermore, in vivo testing of our designed vaccine will be conducted by selecting an appropriate animal model, immunization protocol, and safety evaluation; by performing histopathological examinations. In preclinical studies, effective validation opens the door for clinical trials in humans, eventually leading to the development of a successful vaccine.

## Figures and Tables

**Figure 1 biology-13-00510-f001:**
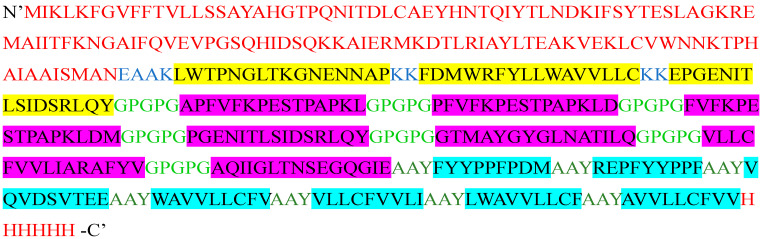
Order of peptides in multivalent vaccine construct. Sequence of the epitopes and linkers was marked by different colors (Aqua colored residues: Cholera enterotoxin subunit B, yellow residues: B-cell epitopes; Pink residues: MHC-II restricted epitopes (HTL); Turquoise residues: MHC-I restricted epitopes (CTL), blue, green and orange residues: linkers.

**Figure 2 biology-13-00510-f002:**
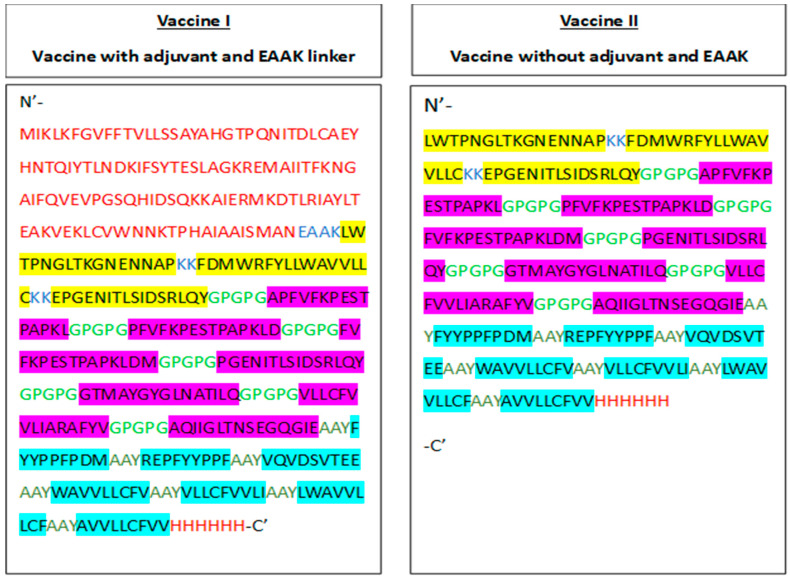
Graphical presentation of Vaccine I and Vaccine II.

**Figure 3 biology-13-00510-f003:**
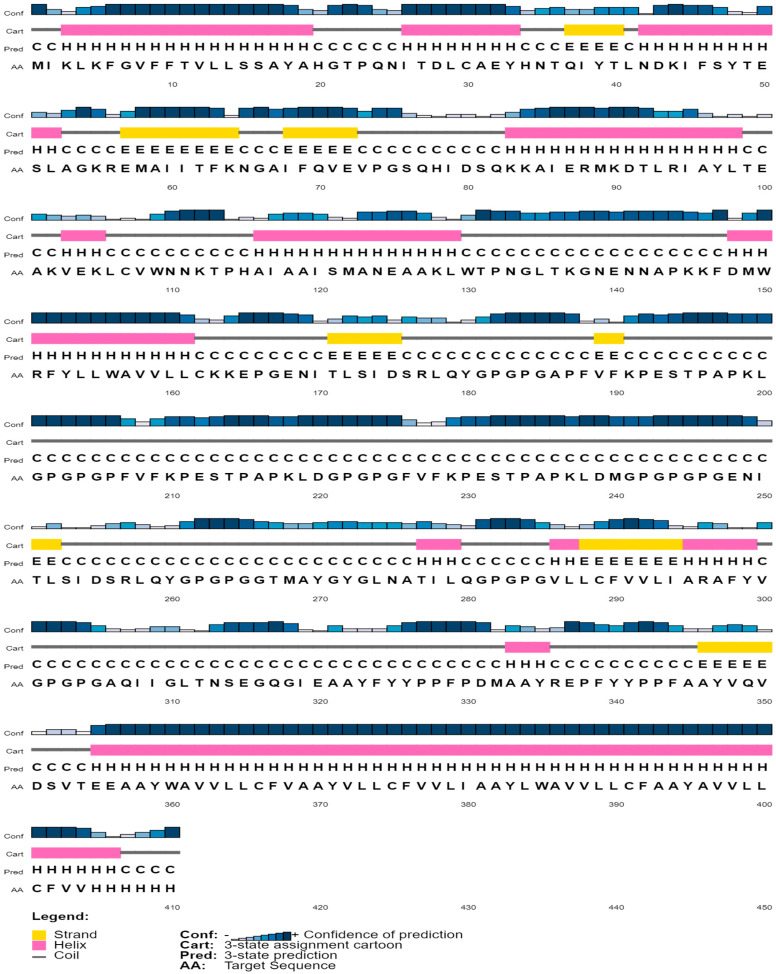
Secondary structure of Vaccine I.

**Figure 4 biology-13-00510-f004:**
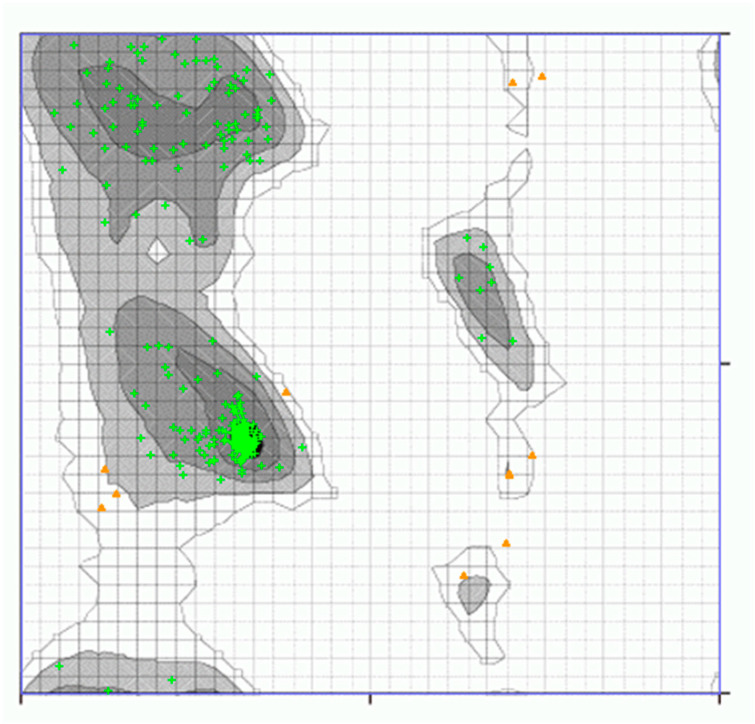
Ramachandaran plot of refined 3D structure (GalaxyWeb) of Vaccine I.

**Figure 5 biology-13-00510-f005:**
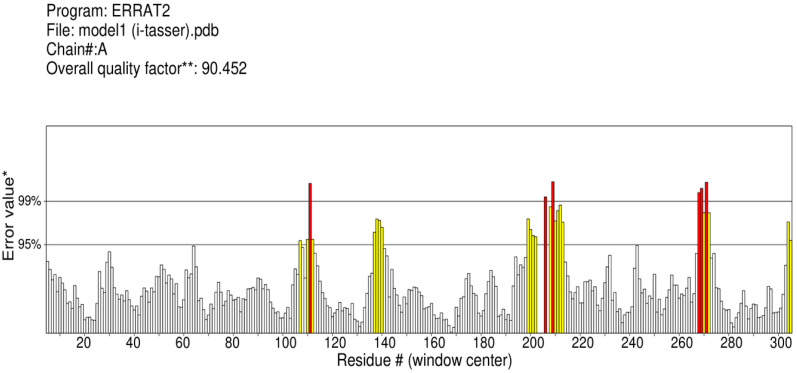
ERRAT plot of 3D structure (iTasser) of Vaccine I before refinement.

**Figure 6 biology-13-00510-f006:**
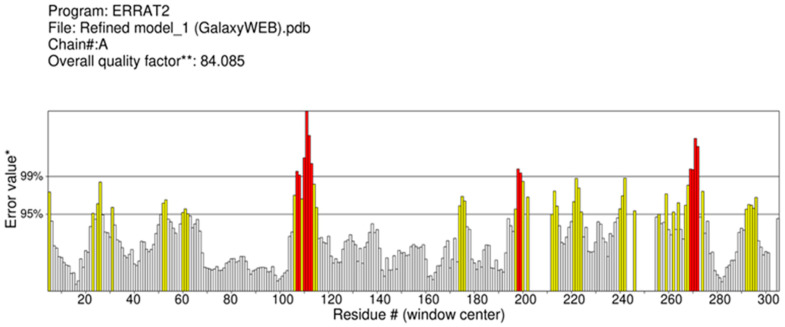
ERRAT Plot of 3D Structure (GalaxyWeb) of Vaccine I after refinement.

**Figure 7 biology-13-00510-f007:**
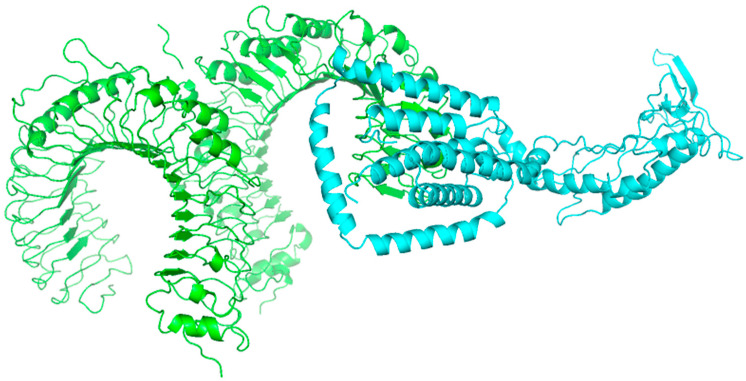
Protein–protein interaction of Vaccine I (ligand: blue) with TLR2 (receptor: green) via ClusPro.

**Figure 8 biology-13-00510-f008:**
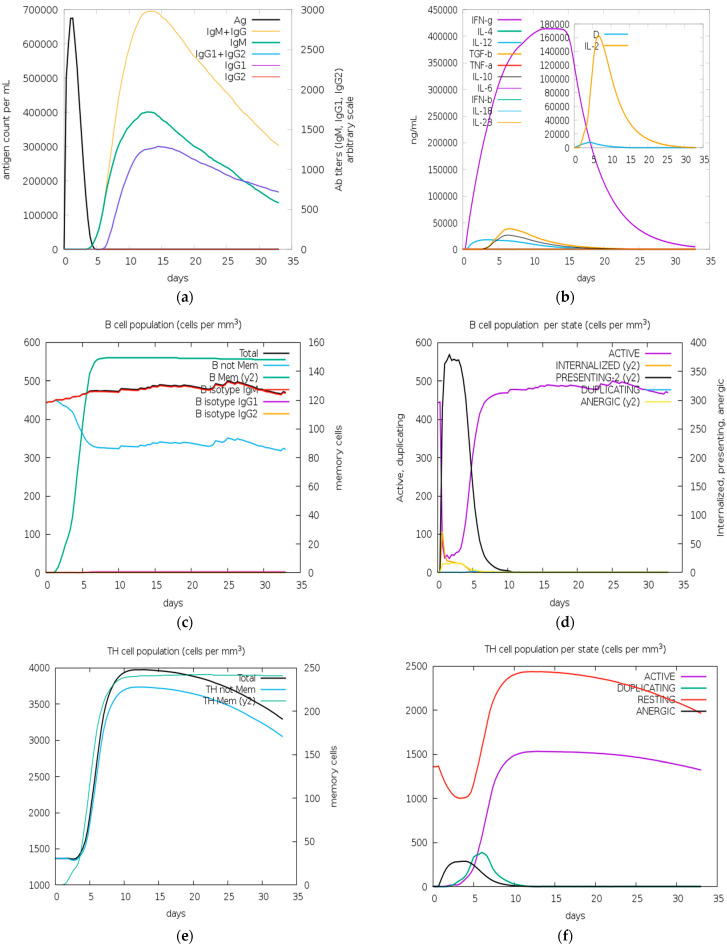
In silico immune simulation of the vaccine with adjuvant (Vaccine I) predicted using the C-IMMSIM tool following injection of vaccine: (**a**) antigen and immunoglobulins: antibodies production by antigen and their sub-division; (**b**) cytokines: rise in the concentration of cytokines and interleukins, D shows the danger signal along with higher production of IL-2 (growth factor); (**c**) total B-cell population count; (**d**) total B-cell population count per state; (**e**) total T-helper cell (CD4) count; (**f**) total population of TH cell per state; (**g**) total population count of T-cytotoxic cell count; (**h**) total TC cell population count per state.

**Figure 9 biology-13-00510-f009:**
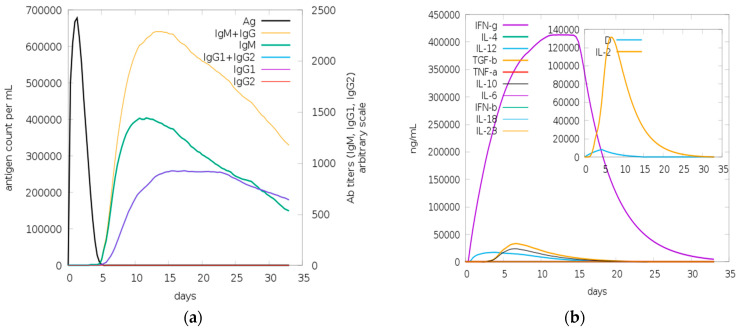
In silico immune simulation of vaccine without adjuvant (Vaccine II) predicted using the C-IMMSIM tool, following injection of vaccine: (**a**) antigen and immunoglobulins: antibodies production by antigen and their sub-division; (**b**) cytokines: rise in the concentration of cytokines and interleukins. D shows the danger signal along with high production of IL-2 (growth factor).

**Figure 10 biology-13-00510-f010:**
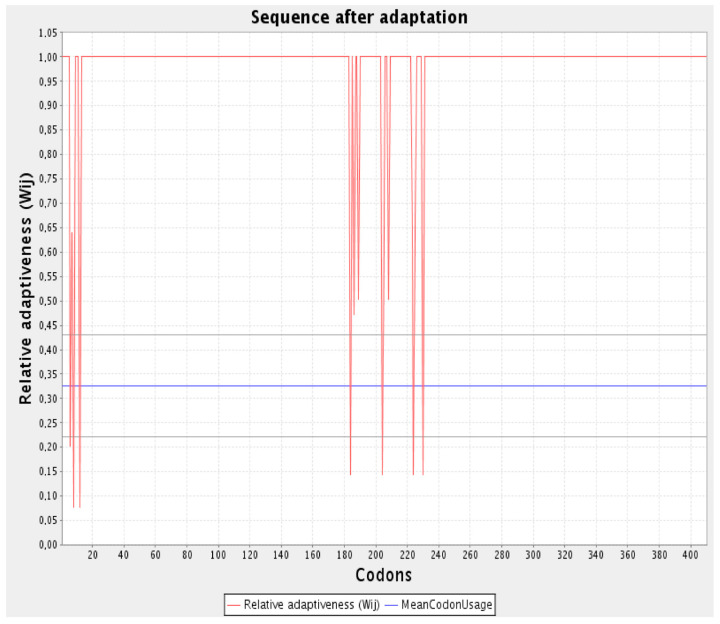
Codon adaptation of Improved DNA.

**Figure 11 biology-13-00510-f011:**
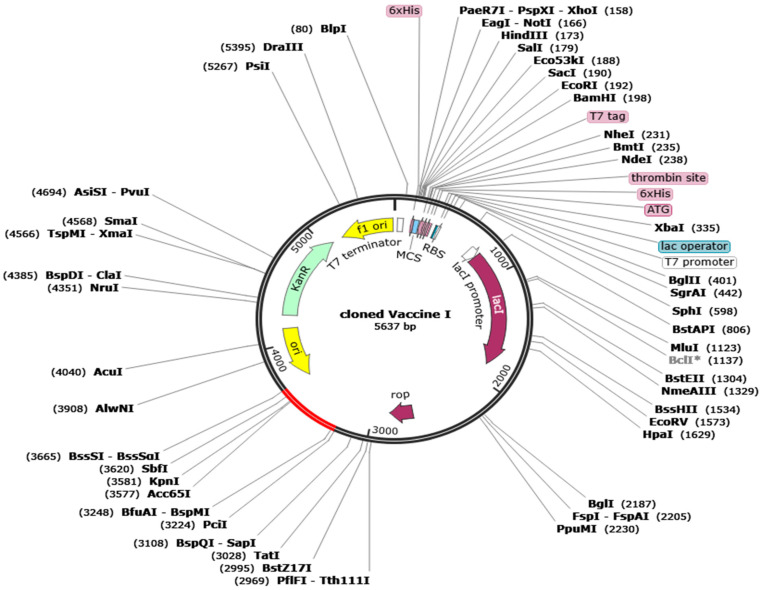
In silico gene cloning of codon optimized vaccine with the expression system *E. coli* K12; pET-30 (+). The vaccine clone was obtained by integrating the vaccine fragment into vector pET-28a (+) of the *E. coli* vector between the restriction sites BssSI (3665) and PciI (3224). Plasmid is shown in black color and the DNA sequence of the vaccine is shown in red color.

**Table 1 biology-13-00510-t001:** Uniprot ID, virulence, and molecular weight, antigenicity, and allergenicity of *S. gordonii* surface proteins.

Sr. #	Gene Name	Protein Name	Uniprot ID	Virulence	MW	Antigenicity	Allergenicity
1	NC01_06050	YSIRK_signal domain protein	A0A0A6S1K7	Yes	53,992.06	0.5118	Non-allergen
2	CC_0349	Peptidase C51 domain-containing protein	Q9AB82	Yes	28,305.13	0.3556	Non-allergen
3	HMPREF9381_0242	AraC family transcriptional regulator	F0HZA2	Yes	34,012.54	0.3336	Non-allergen
4	gtf3 (nss)	Glycosyl transferase	Q9AEU1	Yes	38,007.40	0.3780	Non-allergen
5	ftsI	Peptidoglycan D,D-transpeptidase FtsI	D0CCM7	Yes	67,659.05	0.6184	Non-allergen

**Table 2 biology-13-00510-t002:** Physio-chemical properties and solubility of YSIRK_signal domain protein and peptidoglycan D, D-transpeptidase FtsI.

Sr. #	Property	YSIRK_Signal Domain Protein	Peptidoglycan D,D-Transpeptidase FtsI
3	Molecular weight	53,992.06 kDa	67,659.05 kDa
4	Number of amino acids	469	610
5	Theoretical pI value	9.36	9.78
6	Instability index	44.23	37.64
7	Aliphatic index	44.73	89.69
8	Gravy	−1.277	−0.355
9	Estimated half life	30 h (mammalian reticulocytes, in vitro) >20 h (yeast, in vivo) >10 h (Escherichia coli, in vivo)	30 h (mammalian reticulocytes, in vitro) >20 h (yeast, in vivo) >10 h (Escherichia coli, in vivo)
10	Atomic composition (Carbon, Hydrogen, Nitrogen, Oxygen, Sulfur)	C 2401 H 3676 N 686 O 730 S 5	C 3002 H 4855 N 857 O 877 S 21
11	Total number of negatively charged residues (Asp + Glu)	66	55
12	Total number of positively charged residues (Arg + Lys)	79	81
13	Protein solubility	0.547	0.537

**Table 3 biology-13-00510-t003:** Function and pathway analysis of YSIRK_signal domain protein and peptidoglycan D, D-transpeptidase FtsI.

Sr. #	Protein Name	Uniprot ID	Molecular Function	Biological Function	Pathway
**1**	YSIRK_signal domain protein	A0A0A6S1K7	Carbohydrate binding, Glycopeptides alpha-N-acetylgalactosaminidase activity		No hits
**2**	Peptidoglycan D,D-transpeptidase FtsI	D0CCM7	Penicillin binding, peptidoglycan glycosyltransferase activity, serine-type D-Ala-D-Ala carboxypeptidase activity.	Cell wall organization, division septum assembly, FtsZ-dependent cytokinesis, peptidoglycan biosynthetic process, proteolysis, regulation of cell shape.	Cell wall biogenesis; peptidoglycan biosynthesis.

**Table 4 biology-13-00510-t004:** Antigenicity, allergicity, toxicity, and sub-cellular localization evaluation of B-cell epitopes.

Protein	Sequence/Epitope	Position	Score	Antigenicity	Allergenicity	Toxicity	Sub-Cellular Localization	MW
YSIRK_signal domain-containing protein	LWTPNGLTKGNENNAP	120	0.83	0.4168	Non-allergen	Non-toxic	Outside	1725.88
Peptidoglycan D,D-transpeptidase FtsI	FDMWRFYLLWAVVLLC	24	0.71	1.5824	Non-allergen	Non-toxic	Outside	2075.56
	EPGENITLSIDSRLQY	255	0.62	1.0752	Non-allergen	Non-toxic	Outside	1835

**Table 5 biology-13-00510-t005:** MHC-I alleles binding peptides of YSIRK_signal domain-containing protein and peptidoglycan D,D-transpeptidase FtsI.

Protein	Peptide Sequence/Epitope	MHC-I Alleles	ic50	Antigenicity
YSIRK_signal domain-containing protein	FYYPPFPDM	HLA-A*29:02, HLA-A*23:01, HLA-A*02:06, HLA-A*24:02, HLA-A*30:02, HLA-A*30:01, HLA-A*68:02, HLA-A*31:01, HLA-A*02:01, HLA-A*26:01, HLA-A*25:01, HLA-A*03:01, HLA-A*01:01, HLA-A*11:01, HLA-A*68:01, HLA-A*32:01	63.24	1.9204
	REPFYYPPF	HLA-A*32:01, HLA-A*24:02, HLA-A*23:01, HLA-A*02:06, HLA-A*30:01, HLA-A*29:02, HLA-A*30:02, HLA-A*26:01, HLA-A*31:01, HLA-A*03:01, HLA-A*01:01, HLA-A*25:01, HLA-A*11:01, HLA-A*68:02, HLA-A*02:01, HLA-A*68:01	257.72	1.3015
	VQVDSVTEE	HLA-A*02:06, HLA-A*26:01, HLA-A*11:01, HLA-A*31:01, HLA-A*30:01, HLA-A*02:01, HLA-A*30:02, HLA-A*29:02, HLA-A*01:01, HLA-A*68:01, HLA-A*23:01, HLA-A*25:01, HLA-A*68:02, HLA-A*03:01, HLA-A*32:01, HLA-A*24:02	739.1	1.4495
Peptidoglycan D,D-transpeptidase FtsI	WAVVLLCFV	HLA-A*02:06, HLA-A*68:02, HLA-A*02:01, HLA-A*30:01, HLA-A*31:01, HLA-A*01:01, HLA-A*68:01, HLA-A*30:02, HLA-A*29:02, HLA-A*26:01, HLA-A*23:01, HLA-A*03:01, HLA-A*25:01, HLA-A*11:01, HLA-A*24:02, HLA-A*32:01	14.9	1.2845
	VLLCFVVLI	HLA-A*02:01, HLA-A*02:06, HLA-A*32:01, HLA-A*23:01, HLA-A*30:01, HLA-A*68:02, HLA-A*31:01, HLA-A*24:02, HLA-A*29:02, HLA-A*03:01, HLA-A*01:01, HLA-A*11:01, HLA-A*68:01, HLA-A*30:02, HLA-A*26:01, HLA-A*25:01	21.9	1.5192
	LWAVVLLCF	HLA-A*23:01, HLA-A*24:02, HLA-A*29:02, HLA-A*02:06, HLA-A*30:02, HLA-A*02:01, HLA-A*31:01, HLA-A*30:01, HLA-A*01:01, HLA-A*32:01, HLA-A*03:01, HLA-A*26:01, HLA-A*68:02, HLA-A*25:01, HLA-A*68:01, HLA-A*11:01	45.65	1.5198
	AVVLLCFVV	HLA-A*02:06, HLA-A*02:01, HLA-A*68:02, HLA-A*30:01, HLA-A*31:01, HLA-A*11:01, HLA-A*32:01, HLA-A*30:02, HLA-A*03:01, HLA-A*29:02, HLA-A*26:01, HLA-A*01:01, HLA-A*23:01, HLA-A*24:02, HLA-A*68:01, HLA-A*25:01	80.01	1.4047

**Table 6 biology-13-00510-t006:** MHC-II alleles binding peptides of YSIRK_signal domain-containing protein and peptidoglycan D,D-transpeptidase FtsI.

Protein	Sequence/Epitope	MHC-II Alleles	ic50	Antigenicity
YSIRK_signal domain-containing protein	APFVFKPESTPAPKL	HLA-DRB1*04:01, HLA-DRB1*08:02, HLA-DRB1*16:02, HLA-DRB1*08:01, HLA-DRB1*04:05, HLA-DRB1*01:01, HLA-DRB1*10:01, HLA-DRB1*04:04, HLA-DRB1*11:01, HLA-DRB1*07:01, HLA-DRB1*09:01, HLA-DRB1*13:02, HLA-DRB1*15:01, HLA-DRB1*04:02, HLA-DRB1*13:01, HLA-DRB1*04:03, HLA-DRB1*01:03, HLA-DRB1*03:01, HLA-DRB1*12:01	30.9	1.1214
	PFVFKPESTPAPKLD	HLA-DRB1*04:01, HLA-DRB1*08:02, HLA-DRB1*16:02, HLA-DRB1*08:01, HLA-DRB1*04:05, HLA-DRB1*01:01, HLA-DRB1*10:01, HLA-DRB1*04:04, HLA-DRB1*11:01, HLA-DRB1*07:01, HLA-DRB1*09:01, HLA-DRB1*13:02, HLA-DRB1*15:01, HLA-DRB1*04:02, HLA-DRB1*13:01, HLA-DRB1*01:03, HLA-DRB1*03:01, HLA-DRB1*12:01, HLA-DRB1*04:03	33.1	1.3579
	FVFKPESTPAPKLDM	HLA-DRB1*04:01, HLA-DRB1*08:02, HLA-DRB1*08:01, HLA-DRB1*16:02, HLA-DRB1*01:01, HLA-DRB1*04:05, HLA-DRB1*10:01, HLA-DRB1*04:04, HLA-DRB1*07:01, HLA-DRB1*11:01, HLA-DRB1*09:01, HLA-DRB1*13:02, HLA-DRB1*15:01, HLA-DRB1*04:02, HLA-DRB1*13:01, HLA-DRB1*01:03, HLA-DRB1*12:01, HLA-DRB1*03:01, HLA-DRB1*04:03	39.8	1.4043
Peptidoglycan D,D-transpeptidase FtsI	PGENITLSIDSRLQY	HLA-DRB1*03:01, HLA-DRB1*13:02, HLA-DRB1*09:01, HLA-DRB1*04:04, HLA-DRB1*15:01, HLA-DRB1*04:05, HLA-DRB1*12:01, HLA-DRB1*13:01, HLA-DRB1*07:01, HLA-DRB1*08:02, HLA-DRB1*01:03, HLA-DRB1*01:01, HLA-DRB1*04:01, HLA-DRB1*04:03, HLA-DRB1*11:01, HLA-DRB1*16:02, HLA-DRB1*10:01, HLA-DRB1*08:01, HLA-DRB1*04:02	8.2	1.0722
	GTMAYGYGLNATILQ	HLA-DRB1*09:01, HLA-DRB1*04:01, HLA-DRB1*01:01, HLA-DRB1*04:02, HLA-DRB1*15:01, HLA-DRB1*10:01, HLA-DRB1*04:03, HLA-DRB1*04:05, HLA-DRB1*16:02, HLA-DRB1*13:02, HLA-DRB1*07:01, HLA-DRB1*12:01, HLA-DRB1*08:02, HLA-DRB1*11:01, HLA-DRB1*04:04, HLA-DRB1*08:01, HLA-DRB1*01:03, HLA-DRB1*13:01, HLA-DRB1*03:01	11.8	1.0447
	VLLCFVVLIARAFYV	HLA-DRB1*04:03, HLA-DRB1*12:01, HLA-DRB1*01:03, HLA-DRB1*04:02, HLA-DRB1*07:01, HLA-DRB1*15:01, HLA-DRB1*01:01, HLA-DRB1*08:01, HLA-DRB1*13:01, HLA-DRB1*13:02, HLA-DRB1*11:01, HLA-DRB1*16:02, HLA-DRB1*09:01, HLA-DRB1*08:02, HLA-DRB1*10:01, HLA-DRB1*03:01, HLA-DRB1*04:01, HLA-DRB1*04:04, HLA-DRB1*04:05	26.3	0.9142
	AQIIGLTNSEGQGIE	HLA-DRB1*04:04, HLA-DRB1*04:05, HLA-DRB1*04:01, HLA-DRB1*04:03, HLA-DRB1*08:02, HLA-DRB1*10:01, HLA-DRB1*13:02, HLA-DRB1*07:01, HLA-DRB1*16:02, HLA-DRB1*01:01, HLA-DRB1*04:02, HLA-DRB1*12:01, HLA-DRB1*09:01, HLA-DRB1*15:01, HLA-DRB1*11:01, HLA-DRB1*13:01, HLA-DRB1*08:01, HLA-DRB1*01:03, HLA-DRB1*03:01	30.4	0.9963

**Table 7 biology-13-00510-t007:** Antigenicity, allergenicity, toxicity, and sub-cellular localization of selected MHC-I restricted epitopes.

Protein	Sequence/Epitope	Antigenicity	Allergenicity	Toxicity	Sub-Cellular Localization
YSIRK_signal domain-containing protein	APFVFKPESTPAPKL	1.1214	Non-allergen	Non-toxic	Outside
	PFVFKPESTPAPKLD	1.3579	Non-allergen	Non-toxic	Outside
	FVFKPESTPAPKLDM	1.4043	Non-allergen	Non-toxic	Outside
Peptidoglycan D,D-transpeptidase FtsI	PGENITLSIDSRLQY	1.0722	Non-allergen	Non-toxic	Outside
	GTMAYGYGLNATILQ	1.0447	Non-allergen	Non-toxic	Outside
	VLLCFVVLIARAFYV	0.9142	Non-allergen	Non-toxic	Outside
	AQIIGLTNSEGQGIE	0.9963	Non-allergen	Non-toxic	Outside

**Table 8 biology-13-00510-t008:** Predicted hydrogen bonding of docked Vaccine I with TLR2 via PDBePISA.

Sr. #	Structure 1	Dist. [Å]	Structure 2
1	A: TYR 376 [HH]	1.83	B: SER 309 [O]
2	A: LYS 347 [HZ3]	1.70	B: THR 361 [OG1]
3	A: LYS 347 [HZ1]	1.78	B: THR 363 [OG1]
4	A: GLN 396 [HE22]	2.46	B: SER 409 [OG]
5	A: ASN 345 [OD1]	1.67	B: LYS 385 [HZ2]
6	A: GLU 369 [OE1]	1.81	B: LYS 385 [HZ3]
7	A: GLU 374 [OE1]	2.05	B: ARG 337 [HH21]
8	A: GLU 375 [OE2]	1.79	B: ARG 337 [HH22]
9	A: GLU 375 [OE2]	2.12	B: ARG 337 [HE]

**Table 9 biology-13-00510-t009:** Predicted salt bridges of docked Vaccine I with TLR2 via PDBePISA.

Sr. #	Structure 1	Dist. [Å]	Structure 2
1	A: GLU 369 [OE1]	2.69	B: LYS 387 [NZ]
2	A: GLU 374 [OE1]	2.78	B: ARG 337 [NH2]
3	A: GLU 375 [OE1]	3.26	B: ARG 337 [NH2]
4	A: GLU 375 [OE2]	2.69	B: ARG 337 [NH2]
5	A: GLU 375 [OE2]	2.93	B: ARG 337 [NE]

## Data Availability

Data are contained within the article and Appendix A.

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
