# Peer review of "Reverse Vaccinology Approach to Identify Novel and Immunogenic Targets against *Streptococcus gordonii"

_biology, 2024, doi:10.3390/biology13070510_

Round 1
Reviewer 1 Report
Comments and Suggestions for Authors
In this presented study “Reverse vaccinology approach to identify novel and immunogenic targets against Streptococcus gordonii”, a computational approach was used to construct and design a multi-epitope vaccine (MEV) against S. gordonii.
Abstract is not complete, does not support the significance of the studies, is not organized and structured.
Like it is here: https://www.ncbi.nlm.nih.gov/pmc/articles/PMC8235758/
Or here, this is very similar work that is not referenced:
DOI: 10.1016/j.heliyon.2023.e16148
Introduction lacks review of similar approaches and research done so far; advantages and disadvantages of the approach; examples of methods/approaches used to design similar vaccines.
For publication, the difference to the work that has already been done should be documented.
In the Methods section, some methods are not described.
In the Results section, methods are largely repeated.
There are no discussions, conclusions and further planning provided. For example, the proposed MEV needs to be experimentally validated to ensure its safety and immunogenic profile.
I’d like the authors to address the comments:
12-14: This is not appropriate here and doesn't make sense for the following conclusion. Doesn't emphasize the importance of the given topic.
23-24: “The TLR-2 plays an important role in human immune system.”- this is not relevant here
24-25: “Vaccine I with adjuvant shows higher interactions”- this is not complete result description.
25: “The expression of the protein was determined using in-silico gene cloning.”- This should go up with the method description.
Repetitions:
¾ 32: such as the oral cavity
¾ 33: is mostly found in the oral cavity of humans and animals.
38-39: the sentence needs to be rephrased.
59-60: “As a result, the most effective alternative treatment is a vaccine, which is estimated to save millions of lives each year” – this needs to be referenced.
60-62: “Molecular-omics methods were used to identify immunogenic peptides, including T-cell and B-cell epitopes, and develop a vaccination sequence.”- By whom? Give reference.
63-74: Font deeds to be adjusted.
66-74: The description of the method/approach and the results should be similar here and in the abstract.
92: “virulence was identified using ...”- give more details about the virulence factors (which ones, that could be used further in the discussion).
103-110: this is not method description.
143-147: this is not method description.
179-180: “These tools improved the global and structural quality of tertiary structure [21].”- Improvement should be based on results and documented (clear and concise, in values).
195-198: this is not relevant here.
239-250: This mostly describes the method again.
The same comment for most of the results sections.
254-256: give more details about the virulence factors (which ones, that could be used further in the discussion).
415-416: “… was 0.8562 whereas the vaccine without adjuvant and EAAK linker was 1.0571. This result shows that both the vaccine proteins were antigenic” - The reason and significance of this difference is not discussed further. Is it high or not? ...
419-421: “The result obtained from this tool shows that the vaccine construct with adjuvant and EAAK linker and without adjuvant and EAAK linker were non-allergen.”- to specify the difference in values.
525-526: needs to be rephrased.
527-528: “The result shows vaccine with adjuvant showed better immunogenic response” – to specify the difference in values.
Comments on the Quality of English Language
Minor editing of English language required
Author Response
Reviewer 1
Thank you for your valuable comments and feedback. In the updated manuscript all of the mentioned comments are addressed and described below:
Abstract is not complete, does not support the significance of the studies, is not organized and structured.
It has been revised as an updated manuscript.
Introduction lacks review of similar approaches and research done so far; advantages and disadvantages of the approach; examples of methods/approaches used to design similar vaccines.
Introduction has been improved from line number 66-92.
525-526, 440-442 : needs to be rephrased.
Response:
It has been rephrased in updated manuscript from line number 503-505.
“The result shows vaccine with adjuvant showed better immunogenic response” – to specify the difference in values.
Response:
It has been added to the updated manuscript from line number 507-512. “The result shows vaccine with adjuvant showed better immunogenic response. The black line in the graph represents antigens. When the vaccine is injected, it remains in the body for 5 days as shown in the graph. After 5 days, the body produces antibodies against vaccine. The value of antibodies is very high in body as shown in figure (a) by brown (IgM + IgG), green (IgM), and purple (IgG1) line in graph. So, the high value of antibodies shows that the vaccine response is very good and this will remain in the body for 32 days. It means the vaccine can induce long lasting immunity in the body”.
In the Methods section, some methods are not described.
It has been revised in an updated manuscript and highlighted.
In the Results section, methods are largely repeated.
Thank you for your feedback. Methodology description in results section has been removed and updated.
There are no discussions, conclusions and further planning provided. For example, the proposed MEV needs to be experimentally validated to ensure its safety and immunogenic profile.
It has been added to the conclusion from line number 570-578. “In future, we can check the experimental validity of our proposed vaccine by the expression and purification of the MEV. For that purpose, we can clone the vaccine construct into an appropriate expression vector and induce expression of the vaccine in the host system and then purification of the expressed protein using techniques such as, size exclusion chromatography. The in vitro safety and immunogenicity testing of purified vaccine can be done by number of cytotoxicity assays and immunogenicity assays. Furthermore, in vivo testing of our designed vaccine will be conducted by selecting appropriate animal model, immunization protocol and safety evaluation by performing histopathological examinations. In preclinical studies effective validation open the door for clinical trials in humans, eventually leading to the development of successful vaccine”.
12-14: This is not appropriate here and doesn't make sense for the following conclusion. Doesn't emphasize the importance of the given topic.
It has been added to the updated manuscript in abstract.
“The TLR-2 plays an important role in human immune system.”- this is not relevant here
Response:
This line is removed from abstract.
Vaccine I with adjuvant shows higher interactions”- this is not complete result description.
Response:
It has been rewritten in line number 30-35. Vaccine I with adjuvant shows higher interactions with TLR2 suggested that the vaccine has ability to induce humoral and cell-mediated response to treat and prevent infection.
“The expression of the protein was determined using in-silico gene cloning.”- This should go up with the method description.
Response:
It is moved up with method description.
Repetitions:
such as the oral cavity
is mostly found in the oral cavity of humans and animals.
Response:
Repetition is removed from line 37
38-39: the sentence needs to be rephrased.
Response:
The sentence is rephrased (line 47).
59-60: “As a result, the most effective alternative treatment is a vaccine, which is estimated to save millions of lives each year” – this needs to be referenced.
It has been referenced. Reference number 13.
60-62: “Molecular-omics methods were used to identify immunogenic peptides, including T-cell and B-cell epitopes, and develop a vaccination sequence.”- By whom? Give reference.
It has been referenced. Reference number 14.
63-74: Font needs to be adjusted.
It has been adjusted in the updated manuscript.
66-74: The description of the method/approach and the results should be similar here and in the abstract.
Response:
It has been improved in line number 98-104. Core genes and non-homologous proteins were identified through Subtractive genomics. Essential proteins were predicted from non-homologous proteins. The proteins collected for this study were surface proteins and The antigenic proteins were selected and vaccine was designed on the basis of selected B and T-cell epitopes of the antigenic proteins with the help of linkers and adjuvant. The designed vaccine was docked against TLR-2. The TLR-2 plays an important role in human immune system. Vaccine I with adjuvant shows higher interactions. Then, the expression of the protein was determined using in-silico gene cloning, resulting in a potential MEV construct for S. gordonii.
92: “virulence was identified using ...”- give more details about the virulence factors (which ones, that could be used further in the discussion).
Response:
Description has been added in line number 120-133: The cell wall of S. gordonii is composed of lipoproteins, lipoteichoic acids, repetitive serine-rich adhesins, peptidoglycans, and cell wall proteins that are characterized by individual host receptors. They are involved in virulence and immune regulatory processes that induce inflammatory responses. Dimeric receptors containing TLR2 and TLRx recognize the lipoproteins and LTA. SRR adhesins are important for the binding of S. gordonii to host cells through sialylated glycans. Nucleotide oligomerization domain (NOD), an intracellular receptor, recognized peptidoglycans. Thus, cell wall elements of S.gordonii act as virulent factors that progressively develop diseases with a strong participatory response. The major virulence factors like lipoprotein of S.gordonii, are directly recognized by heterodimers, composed of toll-like receptors TLR2 along with TLR1 or TLR6 on host cell including dental pulp cells, dendritic cells, valve interstitial cells and macrophages. Following activation of TLR2, an adaptor molecule of TLR2, myeloid differentiation primary response 88 (MyD88), mediates the activation of transcription factor that is nuclear factor-kappa B (NF-κB), results in the production of pro-inflammatory cytokines and chemokines, maturation of cell, and infiltration of immune cells into lesions. These processes are involved in inducing inflammatory responses and thus results in development of diseases like apical periodontitis or infective endocarditis
103-110: this is not method description.
Response:
Description has been added in line number 159-164. “T-Cell epitope prediction tool of IEDB analysis resource (http://tools.iedb.org/main/tcell/) was used to evaluate T-cell epitopes. This tool determines the binding affinity of peptides towards the MHC–I and MHC-II based on their IC50 value. On the basis of IC50 score, MHC-I and MHC-II epitopes were selected. For MHC-I binding epitopes (CTL), the prediction method selected was ANN 4.0 and the MHC resource species was human. All the HLA-A* alleles were selected, and the predicted length of epitope was 9-mer”.
143-147: this is not method description.
Response:
Description has been added in line number 201-204: PSIPRED server (http://bioinf.cs.ucl.ac.uk/psipred/&psipred_uuid=7d99d650-1289-11ed-acee-00163e100d53) was used for the secondary structure prediction of vaccine construct. Protein primary sequence is submitted in this tool by selecting predict secondary structure. It will predict the protein secondary structure in the form of coils, helix and sheets. This server shows the results with high accuracy.
179-180: “These tools improved the global and structural quality of tertiary structure [21].”- Improvement should be based on results and documented (clear and concise, in values).
It has been added to the updated manuscript line number 211-215.
195-198: this is not relevant here.
Response:
220: removed
239-250: This mostly describes the method again.The same comment for most of the results sections.
Response:
Method description is removed from results.
Give more details about the virulence factors (which ones, that could be used further in the discussion).
Response:
Description of virulence factors has been added in line number 120-133.
415-416: “… was 0.8562 whereas the vaccine without adjuvant and EAAK linker was 1.0571. This result shows that both the vaccine proteins were antigenic” - The reason and significance of this difference is not discussed further. Is it high or not? ...
Response:
The reason has been mentioned in line number 408-412. “The antigenic score of the protein sequence of vaccine construct with adjuvant and EAAK linker generated by VaxiJen v2.0 was 0.8562 whereas the vaccine without adjuvant and EAAK linker was 1.0571. The antigenic value of vaccine without adjuvant and EAAK linker is high than the vaccine with adjuvant and EAAK linker. This result shows that both the vaccine proteins were antigenic. This means both the vaccines have ability to induce an immune response when injected into the human body”.
419-421: “The result obtained from this tool shows that the vaccine construct with adjuvant and EAAK linker and without adjuvant and EAAK linker were non-allergen.”- to specify the difference in values.
Response:
The antigenic score of the protein sequence of vaccine construct with adjuvant and EAAK linker generated by VaxiJen v2.0 was 0.8562 whereas the vaccine without adjuvant and EAAK linker was 1.0571. AllerTOP v2.0 server was used to predict the allergenic potential of vaccine. The result obtained from this tool didn’t show the values. It only shows whether the vaccine is allergen or non-allergen.

Reviewer 2 Report
Comments and Suggestions for Authors
To Author:
Streptococcus gordonii is a Gram-positive bacterium with a high mortality rate when it infects humans. Although antibiotics can treat Streptococcus gordonii infections, antibiotic resistance is increasing. Therefore, there is a need to find new treatments for Streptococcus gordonii infections. In this study, the authors screened a new target Streptococcus gordonii peptide vaccine, this vaccine can activate B cells and T cells very well. I considered this research article to be significant. However, I have several suggestions before it can be accepted.
Comments:
(1) The authors can improve the clarity and conciseness of the abstract.
(2) There is currently a lack of in vivo animal experiments to verify the effectiveness of selected peptide vaccines in preventing Streptococcus gordonii infection.
(3) The authors should offer a more critical discussion of the study's limitations and challenges for clinical application.
Author Response
Reviewer 2
Thank you for your valuable comments and feedback. In the updated manuscript all of the mentioned comments are addressed and described below:
- The authors can improve the clarity and conciseness of the abstract.
The abstract has been improved in revised manuscript.
(2) There is currently a lack of in vivo animal experiments to verify the effectiveness of selected peptide vaccines in preventing Streptococcus gordonii infection.
Response:
We really acknowledged the reviewer for their valuable comments. Indeed, the study is based on in silico methods and future prospects of this study are in line with the reviewer’s suggestions. However, this manuscript only highlights immunoinformatics approaches used for identification of novel and immunogenic targets. Currently, due to a lack of funding and resources, this study has been restricted to in silico only. Owing to the availability of research funds in the future, we will definitely take this in silico study to experimental validation as per the reviewer’s suggestion.
(3) The authors should offer a more critical discussion of the study's limitations and challenges for clinical application
It has been added in line number 395-400.

Reviewer 3 Report
Comments and Suggestions for Authors
Please see the attached reviewer report.

Comments on the Quality of English Language
Moderate revision is required.
Author Response
Reviewer 3.
The paper has a similarity of 11% with the following thesis, but in the ref list the thesis has not been stated. The similarity must be reduced, and corresponding thesis must be included in the ref list:
Response: The reference is included.
Sajjad, K. (2022). Reverse Vaccinology Approach for Vaccine Development Against Streptococcus agalactiae (Doctoral dissertation, CAPITAL UNIVERSITY).
What is the main strategy in selecting the proteins in the paper.
Response:
Selection of proteins was based on their sub-cellular localization.
Selection criteria for protein selection is mentioned in line number 266-268. Five surface proteins were selected including two extracellular proteins (YSIRK_signal domain protein, and Peptidase C51 domain-containing protein) and three membrane proteins (AraC family transcriptional sregulator, Glycosyl transferase, Peptidoglycan D, D-transpeptidase FtsI) using CELLO tool.
Selection criteria for protein selection is mentioned in line number 277-279. The proteins were selected based on antigenicity and allergenicity. YSIRK_signal domain protein and Peptidoglycan D,D-transpeptidase FtsI were antigenic and non-allergen, so these two proteins were selected for further analysis.
The paper has a number of typos, they must be corrected. For example:
Response:
Typing mistakes are corrected.
Figures must be redrawn so that they can be seen and recognized clearly.
Response:
Figures are redrawn.
Instead of using past simple tense, it is recommended to use simple present one in the abstract.
Response:
It has been updated in revised manuscript.

Round 2
Reviewer 1 Report
Comments and Suggestions for Authors
I’d like the authors to address the comments:
15-20: “Many species of streptococci live in humans without causing symptoms; however, some types can lead to a variety of serious illnesses - from sinus infection to pneumonia. Particularly, infections caused by viridans streptococci worsen when bacteria invade other parts of the body. When bacteria go into the bloodstream, it causes serious infection, called infective endocarditis. These heart infections can be lethal and require treatment.” – this is more than unnecessary here.
24-29: could be written more laconically.
32: needs to be rephrased. It is confusing.
52: “cause of infectious disease called infective endocarditis (IE)” - cause of infective endocarditis (IE).
90-91: “for instance, require substantial validation. To study the immunogenicity of the MEV in 90 an experimental setting, the epitopes must be arranged in the proper order” – needs to be rephrased like this:
For example, extensive validation is required to ensure that the epitopes are arranged in the correct order to study the immunogenicity of the MEV in an experimental setting.
120-121: “Then, the expression of the protein was determined using in-silico gene cloning, resulting in a potential MEV construct for S. gordonii.” – the font size needs to be corrected.
143-156: This is a literature review. Write in two sentences what virulence factors you are trying to identify.
668-669: “Antigenicity and allergenicity of both vaccines suggested that both the vaccines were antigenic and no-allergen” - correct:
Antigenicity and allergenicity of both vaccines suggested that both vaccines were antigenic and non-allergenic.
669-670: “The physiochemical properties and structure of the protein was predicted. The predicted tertiary structure of the protein was validated and refined” - needs to be rephrased like that:
The physicochemical properties and tertiary structure of the protein were predicted. These predictions were validated and refined.
Comments on the Quality of English Language
Minor editing of English language required
Author Response
Reviewer 1 addressed comments:
Thank you for your valuable comments and feedback. In the updated manuscript all of the mentioned comments are addressed and described below:
15-20: “Many species of streptococci live in humans without causing symptoms; however, some types can lead to a variety of serious illnesses - from sinus infection to pneumonia. Particularly, infections caused by viridans streptococci worsen when bacteria invade other parts of the body. When bacteria go into the bloodstream, it causes serious infection, called infective endocarditis. These heart infections can be lethal and require treatment.” – this is more than unnecessary here.
Response:
This text has been removed and added following text from line number 22-25.
Streptococci are the most prevalent inhabitants of oral microbial communities, and are typical oral commensals found in the human oral cavity. These streptococci, along with many other oral microbes, produce multispecies biofilms which can attach to salivary pellicle components and other oral bacteria via adhesin proteins expressed on the cell surface.
24-29: could be written more laconically.
Response:
This has been rephrased from line number 30-35 as mentioned below.
The pangenomics results revealed that out of 2835 pan genes, 1225 are core genes. Out of these1225 core genes, 643 identified as non-homologous proteins by subtractive proteomics. A total of 20 essential proteins are predicted from non-homologous proteins. Among these 20 essential proteins, only 5 are identified as surface proteins. The vaccine construct is designed based on selected B and T-cell epitopes of the antigenic proteins with the help of linkers and adjuvant. The designed vaccine is docked against TLR-2
32: needs to be rephrased. It is confusing.
Response:
It has been rephrased.
52: “cause of infectious disease called infective endocarditis (IE)” - cause of infective endocarditis (IE).
Response:
It has been updated.
90-91: “for instance, require substantial validation. To study the immunogenicity of the MEV in 90 an experimental setting, the epitopes must be arranged in the proper order” – needs to be rephrased like this:
Response:
It has been updated to line number 82-85.
For example, extensive validation is required to ensure that the epitopes are arranged in the correct order to study the immunogenicity of the MEV in an experimental setting.
120-121: “Then, the expression of the protein was determined using in-silico gene cloning, resulting in a potential MEV construct for S. gordonii.” – the font size needs to be corrected.
Response:
It has been corrected.
143-156: This is a literature review. Write in two sentences what virulence factors you are trying to identify.
It has been updated to line number 131-133.
668-669: “Antigenicity and allergenicity of both vaccines suggested that both the vaccines were antigenic and no-allergen” - correct:
Response:
It has been corrected.
Antigenicity and allergenicity of both vaccines suggested that both vaccines were antigenic and non-allergenic.
669-670: “The physiochemical properties and structure of the protein was predicted. The predicted tertiary structure of the protein was validated and refined” - needs to be rephrased like that:
It has been updated to line number 571-572.
The physicochemical properties and tertiary structure of the protein were predicted. These predictions were validated and refined.
